# The Tolerability of the Dengue Vaccine TAK-003 (Qdenga^®^) in German Travelers: The Results of a Prospective Survey

**DOI:** 10.3390/tropicalmed10120352

**Published:** 2025-12-16

**Authors:** Tomas Jelinek, Juliane Kramm, Maik Wagner, Claudia Jelinek

**Affiliations:** Berlin Centre for Travel & Tropical Medicine, Friedrichstrasse 134, 10117 Berlin, Germany

**Keywords:** dengue, live attenuated vaccine, traveler vaccination, adverse event, side effect

## Abstract

Background: The global incidence of dengue has markedly increased over the last decades. Consequently, the risk of infection has increased significantly, resulting in record numbers of imported cases in various European countries and elsewhere in 2024. Methods: In early 2023, TAK-003, a novel, live attenuated vaccine against dengue, became available in Germany. At the Berlin Centre for Travel & Tropical Medicine, we delivered 56,459 doses during the first 24 months of its availability, from February 2023 to February 2025. To obtain data on the tolerability of the vaccine in travelers, an active follow-up survey was initiated. Results: In total, 30,306 (53.7%) vaccinees agreed to being contacted. Of these, 11,827 (39.0%) completed an anonymous questionnaire ≥ 4 weeks after the vaccination. Overall, 6856 (58.0%) were female, and 565 (4.8%) reported having had a prior dengue infection. The average age was 38.3 years (range 4–86 years). An endemic area had been visited by 6309 subjects before answering the questionnaire, and among these, 46 (0.7%) reported a dengue infection while abroad. All cases were mild, and no complications were reported. TAK-003 was applied with other vaccines in 7363 (62.3%) travelers. Local adverse reactions, mostly local pain, were reported by 5263 (47.5%) subjects. Systemic reactions were reported by 4891 subjects (41.4%) and were most frequently fatigue, myalgia, and flu-like symptoms. The majority of adverse events manifested in the second week after vaccination (median 8 days) and were mostly limited to a duration of 1–3 days. A macular exanthema was described by 1844 subjects (15.6%), typically during the second week after the vaccination. Conclusions: Side effects were frequently reported but generally well tolerated. Age groups above 50 years showed a decline in reactogenicity. Co-vaccination was frequent and led to an increase in systemic adverse events. Denominator data of the study population suggest a strong reporting bias towards adverse events. This survey adds evidence of the chronology and characteristics of adverse events associated with TAK-003 and may support decision making when vaccinating dengue-naïve travelers.

## 1. Introduction

The global incidence of dengue has markedly increased over the last few decades. With an estimated 390 million infections annually, the global incidence of dengue escalated from 26.45 million to 58.96 million cases between 1990 and 2021, accompanied by an increase in related deaths from 14,315 to 29,075. Today, dengue is endemic to over 100 countries [1]. Consequently, the risk of infection has increased significantly for travelers. In 2024, record numbers of imported cases were reported in various European countries and elsewhere [2]. In addition, large numbers of travelers with dengue were treated in the endemic destination countries [2].

Dengue presents unique challenges for immunization due to the need for balanced protection against all four serotypes [3]. Primary infection with one serotype typically results in lifelong immunity to that serotype but only transient protection against the others. Subsequent infections with a different serotype tend to increase the risk of severe dengue due to antibody-dependent enhancement [3]. This immunological complexity has made vaccine development particularly difficult.

The first dengue vaccine to be licensed, CYD-TDV (Dengvaxia^®^, Sanofi Pasteur, Lyon, France), is a live attenuated tetravalent vaccine. It has shown variable efficacy depending on prior dengue exposure, age, and serotype. Importantly, CYD-TDV has been associated with an increased risk of severe disease in dengue-naïve individuals, leading to a World Health Organization (WHO) recommendation that it be used only in individuals with confirmed prior dengue infection [4]. Consequently, its use in travelers—who are often dengue-naïve—has been limited and controversial. This vaccine is currently being withdrawn from the market.

More recently, another vaccine has been developed. TAK-003 (QDENGA^®^, Takeda, Osaka, Japan) is an attenuated tetravalent dengue vaccine made from four recombinant dengue viruses (serotypes 1, 3, 4) engineered into a live attenuated dengue serotype 2 virus backbone, teaching the immune system to fight all four virus types [5]. This vaccine has demonstrated favorable efficacy and safety profiles in both seropositive and seronegative individuals, although pivotal studies showed limited efficacy data against DENV-3 and DENV-4 serotypes.

In 2022, TAK-003 received regulatory approval in the European Union for use in individuals aged 4 and older, regardless of prior exposure. The vaccine became available in Germany in February 2023.

For travelers, dengue vaccination offers a potentially valuable preventive tool, particularly for those undertaking long-term or repeated travel to endemic areas. However, implementation requires careful consideration of pre-travel counseling, timing of vaccine administration, and regional epidemiology. The availability of a first new-generation vaccine that is safe and effective in seronegative individuals has the potential to significantly change dengue prevention strategies in the travel medicine setting.

## 2. Methods

In a prospective, self-reported study covering the 24-month period from February 2023 to February 2025, anonymous information on the demographic details and travel destinations of all vaccinees who received TAK-003 was collected from the 10 travel clinics of the BCRT (Berlin Centre and Travel & Tropical Medicine) to obtain denominator data. The travel clinics of the BCRT are spread throughout Germany. In terms of patient visits, the network is among the largest in Europe. As a recruitment measure, all vaccinees who consented were contacted by postal letter 4 weeks after vaccination and asked to access a website to complete an anonymized questionnaire. The commercial online survey tool Questionstar^®^ (Version V843) was used [6]. It enables users to carry out a comprehensive analysis of collected data and to create survey reports that are compatible with all MS Office applications and can be exported and edited as required. The survey included questions on baseline data (age group, sex, place and date of vaccination), travel destination, type, intensity and duration of adverse events, travel after vaccination and diagnosis of dengue infection, and co-vaccinations after the first and second doses of TAK-003. The questionnaire used closed and open questions with the option to report all perceived adverse events. Answers to open questions were grouped for further analysis. To enhance adherence, subjects were not required to complete all questions. All information was self-reported and anonymous, so it was not possible to request further information. This study was approved by the relevant ethics committee (medical board of Berlin).

## 3. Results

TAK-003 (Qdenga^®^) received market authorization from the European Medicines Agency (EMA) in November 2022 and was first distributed in February 2023. Germany was the first country to receive the vaccine, and on 9 February 2023, the Berlin Centre for Travel & Tropical Medicine was the first institution to vaccinate travelers. During the first two years of TAK-003’s availability, 266,519 pre-travel visits occurred at the 10 sites of the BCRT. In this population, 56,459 (21.2%) doses of TAK-003 were administered. In total, 30,077 (53.3%) of the recipients were female, 25,495 (45.2%) male, and 887 (1.6%) classified themselves as being of undefined sex. The age range of vaccinees was 4–86 years, with an average of 38.3 years (median 35 years). When asked about being contacted four weeks later for a follow-up, 30,306 (53.7%) of vaccinees agreed. Of these, 11,827 (39.0%) submitted anonymous information after receiving a letter that asked them to access our study website. These data were further evaluated. Among these, 6856 (58%) subjects were female, 4938 (41.8%) male, and 18 (0.2%) of undefined sex. The age was given in clusters and ranged from 4 to 70+ years with a median of 35 years. A previous dengue infection was reported by 565 (4.8%) subjects. The first dose of TAK-003 was received by 9268 (78.4%) subjects, while the second dose was given to 2521 (21.4%, no answer given by 32); this discrepancy is due to a large proportion of vaccinees receiving their second dose after this study completed data sampling and was closed.

Travel destinations included 181 countries, the 3 most frequently mentioned of which were Thailand (n = 1485), Indonesia (n = 1180), and India (n = 695). Travel duration varied from 1 week to 21 years (the latter given by an emigrant) with a median of 3 weeks.

Before completing the survey, 6309 (53.3%) subjects had already traveled and were thus potentially exposed to dengue infection. Among these, 46 (0.7%) stated that they were laboratory-diagnosed with dengue during their journey. Detailed information on specific laboratory methods is not available; serotypes were not reported. All these dengue episodes were classified as mild by subjects, and no complications or hospitalizations were reported. All 46 subjects reported that they had no dengue prior to vaccination. A total of 6 reported that they received both vaccinations before their journey, whereas 40 received only one.

Local adverse events were reported by 5623 subjects (47.5%) and systemic AEs by 4891 (41.4%) (Table 1). Using the possibility to give open replies, 1294 (10.9%) subjects reported “other” symptoms, with headache (n = 457, 3.9%) being the most frequent in this group. There was no difference between sexes: 48.7% of men reported local adverse events (n = 2406), while 46.9% of women did so (n = 3217) (*p* > 0.05).

Most systemic adverse events appeared in the second week after vaccination (mean onset 9.3 days, median 8 days) and were brief. The majority of adverse events were reported with a duration of 1–3 days (Figure 1). These included local pain (72.2%), local swelling (61.3%), local erythema (47.2%), flu-like symptoms (58.4%), fatigue (59.6%), fever (79.5%), arthralgia (59.0%), and myalgia (64.7%). Slightly longer lasting (up to 7 days) were the majority of exanthemas (76.3%), itching (81.6%), and “others” (71.7%). In terms of severity, most events were classified as very weak, weak, or medium (Figure 2). However, many systemic side effects were classified as strong or very strong: flu-like symptoms in 24.2% and 7.4%, respectively, fatigue in 22.9% and 6.6%, fever in 26.7% and 9.6%, arthralgia in 22.4% and 11.0%, myalgia in 16.8% and 6.9%, exanthema in 25.0% and 13.9%, itching in 14.6% and 6.6%, and “other”—here, primarily headache—in 29.2% and 13.6%.

Five subjects reported side effects that could potentially be classified as severe adverse events:A relapse of a thyroiditis (Hashimoto’s disease) was reported by a female traveler to Panama aged 30–40 years with a planned travel duration of 2 weeks. She had no prior dengue, received her first dengue vaccination, and was co-vaccinated against yellow fever. She stated that she had Hashimoto’s disease for 20 years. With fever after the vaccination, hormonal levels were checked and showed hyperthyroidism for a few days before becoming normal.The first manifestation of hyperthyroidism was reported by a man in the age group 50–60 years who planned to travel to Colombia, had no prior dengue, was not co-vaccinated, and developed hyperthyroidism 6 weeks after being vaccinated.Myocarditis was reported by a 30–40-year-old man who intended to travel to Cambodia for 12 weeks. He had no prior dengue, received his first dengue vaccination and was co-vaccinated against Japanese encephalitis. He reported local pain, erythema, and arthralgia (all medium, 1–3 days); flu-like symptoms, fatigue, and fever (all strong, 8–10 days); and headache (very strong, >30 days). He stated that myocarditis was confirmed using MRI 14 days after vaccination, which resolved over 4 weeks.Sterile meningitis was reported by a woman in the age group 40–50 years who had no prior dengue and received her second dengue vaccination without co-vaccinations for a planned journey to Uganda. About 6 weeks after the vaccination, she developed aseptic meningitis/encephalitis and was briefly hospitalized. Symptoms resolved over 4 weeks.Thrombosis in both legs was reported by a woman in the age group 30–40 years who planned to travel to Barbados. She had no prior dengue, and no co-vaccination was given. Sixteen days after vaccination and still pre-flight, thrombosis in both lower legs was diagnosed. She was hospitalized and treatment was started; the issue was resolved.

Dengue infection prior to vaccination was reported by 564 subjects. Compared to the 11,183 immunologically naïve vaccinees, significant differences were seen with local pain (238 (42.2%) in prior infection vs. 5366 (48.0%) in naïve, *p* = 0.007) and local erythema (114 (20.2%) vs. 3126 (28.0%), *p* < 0.0001), while there were no differences in other adverse events.

When comparing local and systemic adverse events after the first (n = 9237) and second (n = 2523) vaccinations, all were more frequent after the first dose, with the following most significantly so: local pain (4616 (50.0%) after 1st vs. 997 (39.5%) after 2nd, *p* < 0.0001), flu-like symptoms (2699 (29.2%) vs. 263 (10.4%), *p* < 0.0001), fatigue (4290 (46.3%) vs. 594 (23.5%), *p* < 0.0001), fever (1015 (11.0%) vs. 62 (2.5%), *p* < 0.0001), arthralgia (1995 (21.6%) vs. 183 (7.3%), *p* < 0.0001), myalgia (2960 (32.1%) vs. 445 (17.6%), *p* < 0.0001), exanthema (1705 (18.5%) vs. 137 (5.4%), *p* < 0.0001), itching (1222 (13.2%) vs. 246 (9.8%), *p* < 0.0001), and other (1165 (12.6%) vs. 124 (4.9%), *p* < 0.0001).

In total, 4416 subjects (37.3%) received only TAK-003, while between one and four co-vaccinations were given in the majority of 7363 subjects (62.3%), and 48 (0.4%) subjects gave no answer. When comparing subjects with any co-vaccination with those who received none, local pain, flu-like symptoms, fatigue, fever, arthralgia, myalgia, and rash were significantly more frequent in the former, while local swelling and erythema, generalized itching, and other adverse events were not (Table 1). The five most frequent co-vaccinations were rabies (n = 2598), typhoid (n = 1841), Japanese encephalitis (n = 1542), yellow fever (n = 1272), and hepatitis A (n = 681), followed by almost all vaccines licensed for adult use in Germany (Table 2). The only vaccines that were not combined with TAK-003 were routine childhood vaccines and chikungunya, which was not licensed at that time.

When analyzing all adverse events and age groups, reports after the second vaccination were less frequent in all groups (Table 3). The most frequent reports were noted for the first vaccination in the age groups 20–30 and 30–40 years, with a clear tendency to decrease for all types of events in older age groups. Fever was reported in 15.5% of those aged 20–30 years, as compared to 4.5% in the group above 70 years. Similarly, rash was reported most frequently in the 20–30 years group (24.6%), whereas only 3.2% of those above 70 years reported it.

## 4. Discussion

To our knowledge, this is the largest follow-up study on TAK-003 vaccinees in a primarily adult population, and the largest study in travelers. The study results mirror the real-life situation of vaccinating travelers in a professional travel clinic setting. They tend to seek vaccination shortly before travel (in our clinics, 13 days prior to departure on average), are a largely adult population with a high proportion of subjects above 60 years of age, and frequently need several vaccinations before departure, which are given at the same time. All these specifics were not mirrored in the pivotal studies that led to the licensure of TAK-003 [5]. These were performed in endemic areas, and therefore, the study population consisted of children and adolescents. Data on adults were limited with vaccine licensure, and no data were available for subjects above 60 years.

An active surveillance study that recruits from a network with documented patient visits provides a good foundation for denominator data. Prior published work on reactogenicity of TAK-003 in travelers had to rely on anonymous reporting without knowing the underlying numbers of vaccinations [7]. In our setting, we were able to assess the total number of travelers that were counseled during the study period (n = 266,519), the exact number of patients who received a dengue vaccination (n = 56,459), the number of subjects who agreed to being contacted 4 weeks after vaccination (n = 30,306), and the number of those who eventually replied to the anonymous questionnaire (n = 11,827). This loss to follow-up provides a clear indication of the reporting bias that is inherent to real-world studies, and even more so in passive surveillance. A large proportion of vaccinees did not report on the reactogenicity of the vaccines they received, particularly if they experienced no or only limited effects. A common statement of subjects returning to our travel clinics for the second vaccination was that they saw no reason to complete the online questionnaire when side effects were absent or mild. Thus, the percentage of local and systemic side effects reported in this survey is most likely at least double the real percentage in all vaccinees. This aligns with the results of the pivotal studies where reactogenicity was markedly lower than in our setting [5,8].

Most vaccinees were immunologically naïve (95.2%) and received their first vaccination before travel (78.4%). The second dose of TAK-003 was typically given after the journey to provide long-term protection; only 21.4% received both doses before departure. A strength of this study is the extended time interval between vaccination and the survey request—four weeks. Unlike in previous studies [6], this period covered the journey in 53.3% of subjects and provided a robust data set of dengue-naïve travelers who were potentially exposed in endemic areas after receiving the first vaccination. Previously, concerns have been raised about potential complications of dengue infections in vaccinated subjects due to antibody-dependent enhancement [9], leading to very restrictive recommendations in many European countries. In this study, dengue infection during the journey was reported by 46 subjects (0.7% out of those who traveled). All episodes were classified as mild, and no complications or signs of antibody-dependent enhancement were reported. This provides additional evidence of the lack of an ADE risk upon single-dose vaccination with TAK-003 in a dengue-naïve population prior to exposure.

Local and systemic adverse events were frequently reported in our population (47.5% and 41.4%, respectively; Figure 1). Unlike in previous published work [7], there was no difference between men and women. The percentage of reports was higher than in the pivotal studies [5,8] but similar to a previous study in travelers [7]. Manifestation of side effects decreased with age, which is also consistent with previous studies [7]. Except for exanthema, reactions were comparable to previous studies. Exanthema is a systemic reaction that has only rarely been described in pivotal studies [5,8] and not at all in the only prior study in travelers [7], but it occurred in 15.6% of our vaccinees. In clinical practice, this macular rash has been shown to be an important symptom since it initially caused confusion in the affected vaccinees. Like all systemic adverse events, it typically occurred in the second week after vaccination. Subjects did not connect it to the previous vaccination and tended to interpret it as an allergic reaction. This led to consultation with health care providers and, in some cases, presentations at emergency rooms with the risk of receiving antiallergic treatment. Once the symptom had been identified as a common occurrence, vaccination counseling was adapted accordingly. The discrepancy with the pivotal studies [5,8] may be attributed to differences in the study populations. Our study cohort comprised older, more light-skinned and predominantly dengue-naïve vaccinees. In the previously published work in travelers [7], subjects were asked to report one week after vaccination, which may have led to an under-reporting of symptoms that occur later. The longer follow-up interval of 4 weeks after vaccination in our study allowed for these events to be captured.

In general, reported adverse events were short-lived, with a duration of 1–3 days (Figure 2). This aligns with previously published work [5,7,8]. Only exanthema and connected symptoms such as itching tended to last slightly longer (duration of up to 7 days).

Most subjects who reported systemic side effects classified them as weak or medium (Figure 2). However, 21.2% to 38.9% of subjects who experienced systemic reactions classified their symptoms as strong or very strong. This is a higher rate than in pivotal studies [5,8] and might be explained by differences in the study populations. Five subjects reported side effects that might be classified as severe adverse events. Two of these resulted in hospitalization, and all resolved without sequelae. All subjects in this group were below 60 years; medical history and prior medication are largely unknown. As subjects self-reported adverse events through an anonymous database, there is no possibility of verifying reports or following them up. This is clearly a weakness of our study design, which aimed to make reporting as easy as possible. It is possible that reports on strong or very strong symptoms or even potential severe adverse events might have been re-assessed upon physician evaluation. Unlike in randomized controlled trials, the real-world setting of our study lacks a control group. The causality of reported side effects is therefore difficult to interpret.

Prior dengue infection was reported in a subset of our 564 vaccinees, as compared to 11,183 immunologically naïve travelers who received the vaccination. When comparing side effects in these two groups, there was no difference in systemic effects, while local reactions were significantly more frequent in naïve subjects. However, since dengue infections are frequently asymptomatic, differentiating between dengue-naïve individuals and those with a prior dengue infection is difficult, and self-reporting may be misleading.

When comparing the effects of the first and second vaccine doses, all local and systemic side effects decreased with administration of the second dose, most of them significantly. This is in line with previous studies [5,7,8]. Most prior clinical investigations included vaccinees up to 60 years old, whereas this survey included participants aged 4–86 years (Table 3), thus adding safety data for older travelers who are considered at higher risk of dengue infection [10]. Due to the absence of individuals over the age of 60 in pivotal studies, a comparison could not be made. Our findings show peaks for all adverse events after the first and second doses for the age group 20–40 years with a decline with further age. This factor was not shown in pivotal studies since they included only young subjects [5,8] but an indication of this trend was already shown in a previous study in German travelers [7]. This result does not justify a contraindication for the vaccine in elderly travelers [3], particularly given that older patients have a more severe course of dengue.

Upon licensure, data on co-vaccination with yellow fever, hepatitis A, and HPV vaccines were published [11,12], demonstrating proof of principle for live attenuated and inactivated co-vaccination. The actual situation in a travel clinic with a combination of up to four vaccines in one session could only be assessed and followed up in a real-world situation. Co-administration is a common vaccination practice due to time and cost savings. It has also been shown to be well tolerated [13]. In our data set, 62.3% of vaccinees received TAK-003 in combination with at least one other vaccine (Table 1). Some adverse events increased significantly with co-vaccination. This is in line with previous studies on this topic [13,14] which showed similar increases in co-vaccinations. Previous work has also shown that an increase in local and systemic side effects with co-vaccination does not lead to increased incapacitation or reduced routine activities [13]. Due to the anonymous nature of reporting, this detail was not followed up in our study. TAK-003 was most frequently co-administered in combination with rabies, typhoid fever, Japanese encephalitis, or yellow fever vaccine (Table 2). Specific symptoms tended to increase with specific vaccine combinations; e.g., myalgia was more common after co-vaccination with TdapP (40.7% vs. 23.9%). The only prior study on TAK-003 in travelers showed no differences in the incidence of adverse events, but the results may have been affected by a comparatively small number of study participants [7]. However, it must be acknowledged that adverse events related to specific combinations of vaccines are difficult to analyze due to the lack of a control group not receiving TAK-003. This makes it difficult to assess whether observed reactions were primarily due to the combination of vaccines or specific to TAK-003. Moreover, data on the influence of co-morbidities such as diabetes, hypertension, autoimmune diseases, and malignancies in the vaccinees was not available, so these factors could not be tested for correlations with vaccine outcomes and adverse events. This is a major limitation of our data set and may be addressed in future studies.

This study was based on a single voluntary survey conducted 4 weeks after vaccination. We are confident that it identified early- and late-onset adverse events. To make reporting as easy as possible for subjects, surveys were administered through a website for rapid anonymous reporting. These data cannot be directly linked to individual subjects and are impossible to verify. Respondent bias, always an issue when collecting real-world data, was detectable in this study since denominator data were available. Participants reporting adverse events were more likely to answer the questionnaire, thus creating a bias towards higher reactogenicity. Moreover, severity categorization was the subjective perception of the subjects, which probably led to a bias towards strong and very strong side effects. Furthermore, the self-reporting of fever may result in limited accuracy when compared to phase 3 trials. It is challenging to attribute reported adverse events to the vaccine and to compare them against potential baseline reactions in an unvaccinated control group since a placebo group is lacking. Overall, the results of this study demonstrate that adverse events are frequently reported following the administration of TAK-003, but most cases are not severe and do not pose a risk to travelers. There was also no indication of safety issues in vaccinated travelers during their journey to endemic areas.

## Figures and Tables

**Figure 1 tropicalmed-10-00352-f001:**
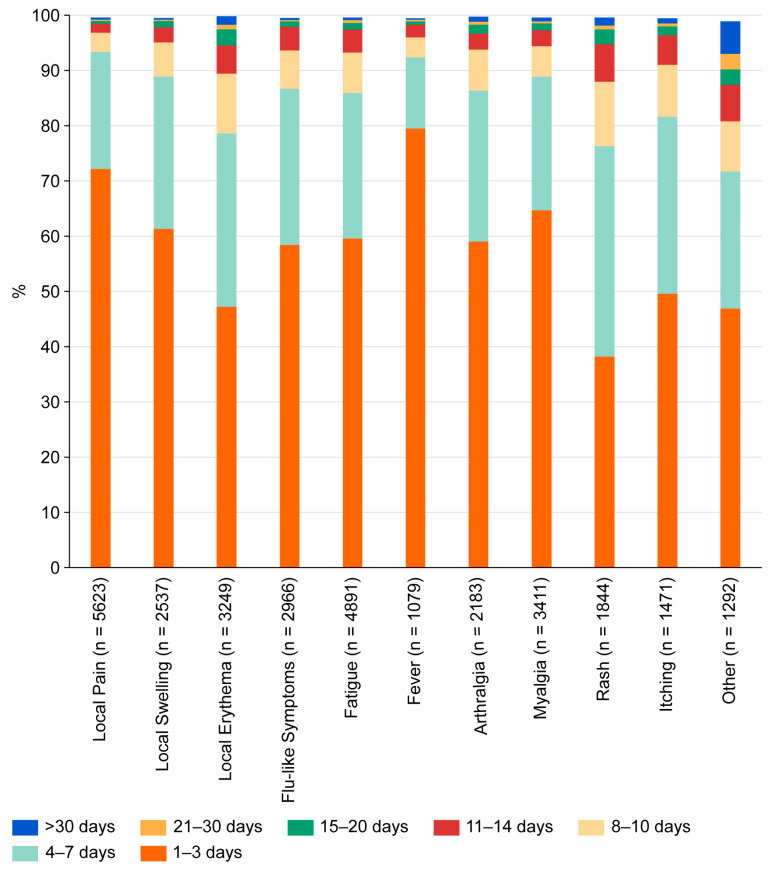
Duration of adverse events reported by subjects after vaccination with TAK-003.

**Figure 2 tropicalmed-10-00352-f002:**
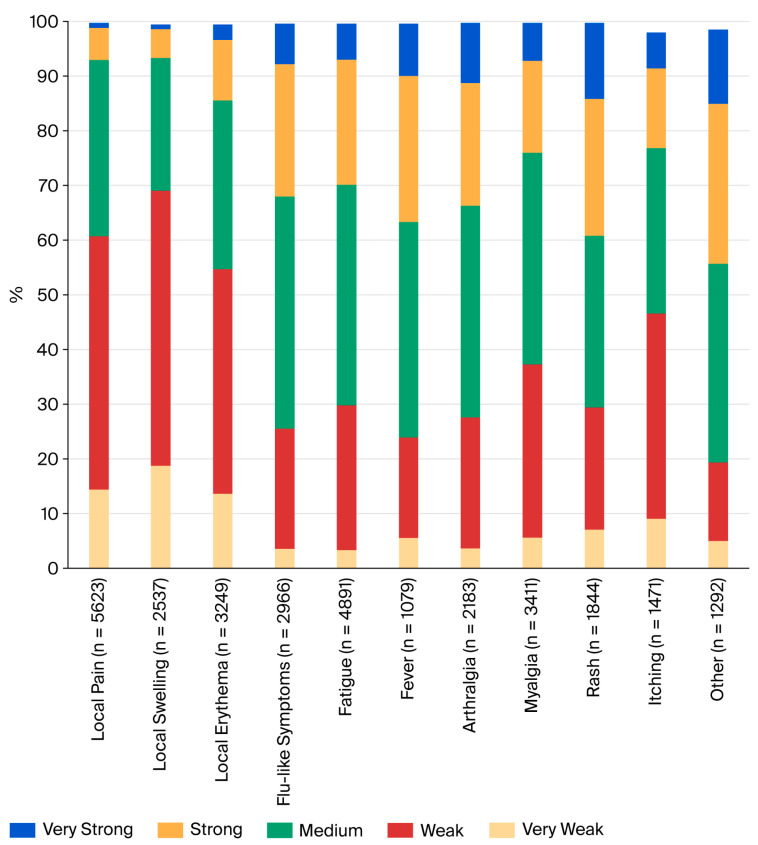
Severity of adverse events reported by subjects after vaccination with TAK-003.

**Table 1 tropicalmed-10-00352-t001:** Adverse events and co-vaccination.

AEs	Entire Group n = 11,827	Any Co-Vaccination n = 7342	No Co-Vaccination n = 4402	*p*-Value
Local Pain	5623 (47.5%)	3815 (52.0%)	1787 (40.6%)	<0.01
Local Swelling	2537 (21.5%)	1635 (21.3%)	889 (20.2%)	0.008
Local Erythema	3249 (27.5%)	2065 (28.1%)	1171 (26.6%)	n.s.
Flu-like Symptoms	2966 (25.1%)	2025 (27.6%)	932 (21.2%)	<0.01
Fatigue	4891 (41.4%)	3351 (45.6%)	1522 (34.6%)	<0.01
Fever > 38.5 °C	1079 (9.1%)	780 (10.6%)	297 (6.8%)	<0.01
Arthralgia	2183 (18.5%)	1504 (20.5%)	674 (15.3%)	<0.01
Myalgia	3411 (28.8%)	2366 (32.2%)	1036 (23.5%)	<0.01
Rash	1844 (15.6%)	1258 (17.1%)	583 (13.2%)	<0.01
Itching	1471 (12.4)	938 (12.8%)	529 (12.0%)	n.s.
Other	457 (3.9%)	862 (11.7%)	425 (9.7%)	0.0007

**Table 2 tropicalmed-10-00352-t002:** Adverse events and co-vaccination.

	Co-Vaccination												
AEs	None N = 1892	Cholera N = 221	YF N = 644	Flu N = 107	Hep A N = 322	Hep A + B N = 173	Jap. Enc. N = 745	MMR N = 46	Men ACWY N = 156	Polio N = 117	TdapP N = 135	Rabies N = 1518	Typhoid N = 894
Flu-like Symptoms	392 (20.7%)	45 (20.4%)	122 (18.9%)	27 (25.2%)	72 (22.4%)	29 (16.8%)	214 (28.7%)	8 (17.4%)	49 (31.4%)	30 (25.6%)	49 (36.3%)	423 (27.9%)	229 (25.6%)
Fatigue	639 (33.8%)	87 (39.4%)	250 (38.8%)	40 (37.4%)	138 (42.9%)	58 (33.5%)	363 (48.7%)	16 (34.8%)	74 (47.4%)	49 (41.9%)	68 (50.4%)	728 (48.0%)	395 (44.2%)
Fever > 38.5 °C	128 (6.8%)	17 (7.7%)	38 (5.9%)	9 (8.4%)	25 (7.8%)	8 (4.6%)	89 (11.9%)	1 (2.2%)	15 (9.6%)	8 (6.8%)	16 (11.9%)	166 (10.9%)	87 (9.7%)
Arthralgia	279 (14.7%)	36 (16.3%)	96 (14.9%)	17 (15.9%)	51 (15.8%)	20 (11.6%)	161 (21.6%)	5 (10.9%)	37 (23.7%)	15 (12.8%)	29 (21.3%)	314 (20.7%)	160 (17.9%)
Myalgia	452 (23.9%)	57 (26.0%)	166 (25.8%)	32 (29.9%)	106 (32.9%)	51 (29.5%)	254 (34.1%)	16 (34.8%)	53 (33.9%)	30 (25.6%)	55 (40.7%)	501 (33%)	284 (31.8%)
Rash	278 (14.7%)	38 (17.2%)	57 (8.9%)	18 (16.8%)	48 (14.9%)	22 (12.7%)	139 (18.7%)	3 (6.5%)	34 (21.8%)	23 (19.7%)	31 (23.0%)	321 (21.1%)	169 (18.9%)
Itching	229 (12.1%)	23 (10.4%)	64 (9.9%)	7 (6.5%)	45 (14.0%)	21 (12.1%)	101 (13.6%)	5 (10.9%)	26 (16.7%)	16 (13.7%)	18 (13.3%)	194 (12.8%)	113 (12.6%)
Other	211 (11.6%)	19 (8.6%)	57 (8.9%)	9 (8.4%)	41 (12.7%)	17 (9.8%)	92 (12.4%)	4 (8.7%)	27 (17.3%)	12 (10.3%)	20 (14.8%)	191 (12.6%)	86 (9.6%)

YF = Yellow Fever, Flu = Influenza, Hep A = Hepatitis A, Hep A + B = Hepatitis A and B, Jap. Enc. = Japanese Encephalitis, MMR = Measles, Mumps, Rubella, Men ACWY = Meningococcal Meningitis Serotypes ACWY, Polio = Poliomyelitis, and TdapP = Tetanus, Diphtheria, Pertussis, Polio.

**Table 3 tropicalmed-10-00352-t003:** Adverse events and age.

Age		4–10 Years		10–20 Years		20–30 Years		30–40 Years		40–50 Years		50–60 Years		60–70 Years		70+ Years	
Vaccination (Total Number Receiving Either 1st or 2nd Dose)		1st N = 41	2nd N = 9	1st N = 378	2nd N = 67	1st N = 1926	2nd N = 312	1st N = 2130	2nd N = 457	1st N = 1177	2nd N = 367	1st N = 1607	2nd N = 603	1st N = 933	2nd N = 390	1st N = 157	2nd N = 69
**Adverse Events**	**Local Pain**	28 (68.3%)	1 (11.1%)	207 (54.8%)	27 (40.3%)	1150 (59.7%)	170 (54.5%)	1174 (55.1%)	227 (49.7%)	570 (48.4%)	147 (40.1%)	642 (40.0%)	193 (32.0%)	355 (38.1%)	126 (32.3%)	34 (21.7%)	16 (23.2%)
	**Local Swelling**	10 (24.4%)	1 (11.1%)	75 (19.8%)	9 (13.4%)	446 (23.2%)	82 (26.3%)	466 (21.9%)	109 (23.9%)	260 (22.1%)	80 (21.8%)	326 (20.3%)	118 (19.6%)	200 (21.4%)	74 (19.0%)	20 (12.7%)	10 (14.5%)
	**Local Erythema**	12 (29.3%)	0 (0%)	94 (24.9%)	15 (22.4%)	569 (29.5%)	95 (30.5%)	617 (29.0%)	133 (29.1%)	341 (29.0%)	95 (25.9%)	460 (28.6%)	147 (24.4%)	215 (23.0%)	92 (23.6%)	27 (17.2%)	15 (21.7%)
	**Flu-like Symptoms**	7 (17.1%)	1 (11.1%)	84 (22.2%)	6 (9.0%)	692 (35.9%)	51 (16.4%)	701 (32.9%)	49 (10.7%)	350 (29.7%)	36 (9.8%)	405 (25.2%)	49 (8.1%)	197 (21.1%)	39 (10.0%)	18 (11.5%)	2 (2.9%)
	**Fatigue**	12 (29.3%)	2 (22.2%)	174 (46.0%)	18 (26.9%)	1112 (57.7%)	104 (33.3%)	1103 (51.8%)	127 (27.8%)	536 (45.5%)	91 (24.8%)	597 (37.2%)	105 (17.4%)	308 (33.0%)	71 (18.2%)	29 (18.5%)	8 (11.6%)
	**Fever > 38.5 °C**	4 (9.8%)	1 (11.1%)	41 (10.9%)	3 (4.5%)	294 (15.3%)	13 (4.2%)	266 (12.5%)	11 (2.4%)	131 (11.1%)	5 (1.4%)	130 (8.1%)	12 (2.0%)	40 (4.3%)	8 (2.1%)	7 (4.5%)	2 (2.9%)
	**Arthralgia**	4 (9.8%)	0 (0%)	65 (17.2%)	7 (10.5%)	506 (26.3%)	31 (9.9%)	513 (24.1%)	37 (8.1%)	253 (21.5%)	18 (4.9%)	289 (18.0%)	36 (6.0%)	150 (16.1%)	20 (5.1%)	14 (8.9%)	3 (4.4%)
	**Myalgia**	9 (22.0%)	0 (0%)	119 (31.5%)	17 (25.4%)	755 (39.2%)	69 (22.1%)	755 (35.5%)	93 (20.4%)	361 (30.7%)	52 (14.2%)	436 (27.1%)	90 (14.9%)	218 (23.4%)	61 (15.6%)	21 (13.4%)	5 (7.3%)
	**Rash**	9 (22.0%)	0 (0%)	64 (16.9%)	7 (10.5%)	474 (24.6%)	25 (8.0%)	476 (22.4%)	31 (6.8%)	241 (12.3%)	19 (5.2%)	214 (13.3%)	23 (3.8%)	68 (7.3%)	11 (2.8%)	5 (3.2%)	1 (1.5%)
	**Itching**	7 (17.1%)	0 (0%)	56 (14.8%)	8 (11.9%)	344 (17.9%)	38 (12.2%)	313 (14.7%)	52 (11.4%)	145 (12.3%)	36 (9.8%)	169 (10.5%)	58 (9.6%)	72 (7.7%)	21 (5.4%)	10 (6.4%)	3 (4.4%)
	**Other**	3 (7.3%)	0 (0%)	38 (10.1%)	3 (4.5%)	296 (15.4%)	13 (4.2%)	302 (14.2%)	45 (9.9%)	153 (13.0%)	17 (4.6%)	171 (10.6%)	14 (2.3%)	72 (7.7%)	15 (3.8%)	9 (5.7%)	1 (1.5%)

## Data Availability

The original contributions presented in this study are included in the article. Further inquiries can be directed to the corresponding author.

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
