# Peer review of "The Tolerability of the Dengue Vaccine TAK-003 (Qdenga^®^) in German Travelers: The Results of a Prospective Survey"

_tropicalmed, 2025, doi:10.3390/tropicalmed10120352_

Round 1
Reviewer 1 Report
Comments and Suggestions for Authors
The paper entitled Tolerability of the Dengue Vaccine TAK003 (Qdenga®) in German Travellers: Results of a Prospective Survey by Tomas Jelinek et al; is a timely observational study with practical relevance for travel medicine, but clarity and methodological detail need improvement. I have the following suggestions and comments for improvement.
- Please specify that this was a prospective, self-reported study, and clarify recruitment methods, inclusion/exclusion criteria, and what “active follow-up” entailed. Indicate whether the survey was validated and how open-ended responses were analyzed.
- I suggest including confidence intervals or standard deviations for key metrics. Clarify whether post-travel dengue cases were lab-confirmed and whether serotypes were identified. Given the 39% response rate, a brief note on non-response bias would be helpful.
- Please highlight novel aspects such as the timing of adverse events and co-vaccination effects. Discuss reporting bias and its impact on reactogenicity estimates. The comparison with pivotal trials is useful; consider adding implications for older adults.
- Emphasize the clinical relevance of mild dengue cases post-vaccination, especially regarding ADE. Note how follow-up timing may have improved detection of delayed events and underscore the value of age-specific safety data.
- I also suggest acknowledging limitations of anonymous self-reporting and the absence of a control group when interpreting causality.
It is well written. Minor corrections suggested.
Author Response
We wish to thanks the reviewers for their helpful and insghtful comments. We account for them as follows:
- Please specify that this was a prospective, self-reported study, and clarify recruitment methods, inclusion/exclusion criteria, and what “active follow-up” entailed. Indicate whether the survey was validated and how open-ended responses were analyzed.
The methods section has been updated stating that this was a prospective, self-reported study. Recruitment methods and inclusion/exclusion criteria, active follow-up were clarified. Open-ended responses were grouped for further analysis, this information has also been added to the methods section. - I suggest including confidence intervals or standard deviations for key metrics. Clarify whether post-travel dengue cases were lab-confirmed and whether serotypes were identified. Given the 39% response rate, a brief note on non-response bias would be helpful.
Information on post travel cases was given by anonymous reporting. Subjects stated that their diagnosis was laboratory confirmed. However, no detailed information is available on lab methods. This information has been added to the results section. The obvious response bias is a weakness of this type of study. This is stated broadly in the discussion, and has been elaborated following the suggestion of the reviewer. - Please highlight novel aspects such as the timing of adverse events and co-vaccination effects. Discuss reporting bias and its impact on reactogenicity estimates. The comparison with pivotal trials is useful; consider adding implications for older adults.
Novel apsects are pointed out in the discussion, in particular the benefits of delaying reporting of adverse events in order to catch late onset. Results on co-vaccination are also shown in the resuts section and are elaborated in the discussion. - Emphasize the clinical relevance of mild dengue cases post-vaccination, especially regarding ADE. Note how follow-up timing may have improved detection of delayed events and underscore the value of age-specific safety data.
The clinical relevance of mild dengue cases post-vaccination is elaborated in the discussion, as well as the benefits of delaying reporting of adverse events in order to catch late onset. - I also suggest acknowledging limitations of anonymous self-reporting and the absence of a control group when interpreting causality.
Statements on the imitations of anonymous self-reporting and the absence of a control group when interpreting causality have been added to the discussion.
Reviewer 2 Report
Comments and Suggestions for Authors
This study is adding data on the safety and tolerability of TAK003 vaccine in a German cohort. It also provides the tolerability of the vaccine with the other routine vaccines. Covers a larger cohort, including diverse age groups and provides clear data on the adverse events experienced by the vaccinees. I suggest few minor revisions which are as follows
typo graphical error in lines 85 - the option
line 96- Out of these
Whether ethical approval was obtained to conduct this study?. Please mention that in your methods section.
The authors could have provided some scientific data about the vaccine and its content in the introduction, so that the naive readers will have some idea about the vaccine.
Although, the intensity of the adverse events were less in the people who have received the second dose, the probable mechanism behind the milder manifestations of adverse events could be discussed.
the authors could give an exclusive paragraph on the adverse events experienced by the naive vaccinees, recieving only the TAK003 vaccine but not the other vaccines. As this would throw light on the adverse events exclusively brought about by this vaccine.
It was not clear that the 47 vaccinated people who developed dengue recieved only one dose or 2 doses of TAK003 and whether it is a prmary or secondary dengue?
Also data on the influence of the co-morbidities such as Diabetes, hypertension, autoimmune diseases and malignancies in the vaccinees was not correlated with the vaccine outcomes and adverse events. This is a major limitation of your study.
A paragraph depicting the future directions on the other scopes that were not / could not be covered by your study will help the researchers for future studies.
Author Response
We thank the reviewer for the helpful and insightful comments and account for them as follows:
typo graphical error in lines 85 - the option
-has been corrected
line 96- Out of these
-has been corrected
Whether ethical approval was obtained to conduct this study?. Please mention that in your methods section.
-this is mentioned in the methods section (last sentence)
The authors could have provided some scientific data about the vaccine and its content in the introduction, so that the naive readers will have some idea about the vaccine.
-brief information on the vaccine has been added to the introduction
Although, the intensity of the adverse events were less in the people who have received the second dose, the probable mechanism behind the milder manifestations of adverse events could be discussed.
-this is commonly seen in live attenuated vaccines since the already active immune reaction to the first vaccine dose reduces replication of the second dose.This is stated in the discussion.
the authors could give an exclusive paragraph on the adverse events experienced by the naive vaccinees, recieving only the TAK003 vaccine but not the other vaccines. As this would throw light on the adverse events exclusively brought about by this vaccine.
-this information is shown in detail in table 2 and also outlined in the results section
It was not clear that the 47 vaccinated people who developed dengue recieved only one dose or 2 doses of TAK003 and whether it is a prmary or secondary dengue?
-In our study 46 subjects reported a dengue diagnosis during travel. As reporting was anonymous, we are unable to follow up for further details. However, all 46 subjects reported that they had no dengue prior to vaccination. 6 reported that they received both vaccinations before their journey, 40 received only one. This information has been added to the results section.
Also data on the influence of the co-morbidities such as Diabetes, hypertension, autoimmune diseases and malignancies in the vaccinees was not correlated with the vaccine outcomes and adverse events. This is a major limitation of your study.
- this information is missing in our data set and we agree with the reviewer that this is a major limitation of the study. In order to make reported as easy and fast as possible, we chose to omitt questions on chronic diseases and medication.
A paragraph depicting the future directions on the other scopes that were not / could not be covered by your study will help the researchers for future studies.
-a respective paragraph has been added to the discussion, on particular stating the lack of data on co-morbidities.
Reviewer 3 Report
Comments and Suggestions for Authors
Tolerability of the Dengue Vaccine TAK003 (Qdenga ® ) in German Travellers: Results of a Prospective Survey
Review
The study by Jelinek et al provides a brief account of the safety of the TAK003 vaccine in a large cohort of travellers from Germany. After WHO’s recommendation for use, the vaccine has been made available for travellers since 2023 in Europe. The data is self-reported by the travellers after 4 weeks of administration. The vaccine shows some adverse events which are short lived, with no serious complications. The study adds value in imparting the timely information regarding a majority naïve population and their response to vaccination.
- Although the study is insightful, the data is presented rather superficially. There is a lack of depth in the data. The data is not represented well in terms of figures. No statistical models have been applied to gain more insight in the data. In the individuals where there had been a pre-exposure to dengue or the ones with adverse events could have been further followed up to gain more insights on those particular cases.
- The study characteristics could have been represented in the form of a flow chart, explaining the intricacies in a better manner and the percentages of how many subjects are included from what group, their inclusion and exclusion criteria would have been represented in a coherent manner.
- The introduction does not deal with the serotype specific protection TAK003 provides and does not point out this aspect of the vaccine. Its constitution, efficacy and safety are not well described and referenced.
- Table 2 and 3 could have been made into figures to represent the data better.
- Some statistical analysis and confidence values could have been provided to better understand the data, all in terms of days of symptoms, their range differences in age and symptoms due to co-vaccination.
- The authors should have gotten more information on the individuals that got dengue after vaccination. Although, its understandable that the study design does not allow to do so, but it was valuable information which should not have been ignored.
Line 15-20 The information and numbers could be distilled further to make the abstract concise and crisp.
Line 43 Please provide reference.
Line 44 Please provide reference.
Line 52-54 Please provide reference.
Line 60 Seems like this is the only other vaccine that has been developed.
Line 69 No information about limited safety against DENV-3 and DENV-4 serotypes.
Line 98 The ages are somewhere written as average as well as median and the values don’t match in Abstract and Results.
Line 116 Table 1 not described here but later, if that order is appropriate then change the order of the tables.
Line 165 The number does not match with the one in abstract. Also, the observation about these individuals not mentioned in the abstract.
Line 233 The number of incidences of dengue post vaccination does not match the value written in abstract and results.

Author Response
We wish to thank the revierwer for the helpful and insightful comments and account for them as follows:
- Although the study is insightful, the data is presented rather superficially. There is a lack of depth in the data. The data is not represented well in terms of figures. No statistical models have been applied to gain more insight in the data. In the individuals where there had been a pre-exposure to dengue or the ones with adverse events could have been further followed up to gain more insights on those particular cases.
We wish to point out that this is the result of an anonymised data collection. It does not offer the same depth for statistical analysis as a RCT. However, statistical methods were applied where feasible, significant differences are shown throughout the paper.
- The study characteristics could have been represented in the form of a flow chart, explaining the intricacies in a better manner and the percentages of how many subjects are included from what group, their inclusion and exclusion criteria would have been represented in a coherent manner.
This data set is the result of an anonymised reporting. Subjects were not included in groups, as outlined in the methods section. Recruitment was really very straight forward: everybodes who received the vaccine was asked if he/she agreed to being contacted 4 weeks later. If they agreed, the received a letter asking them to access an anonymised questionnaire. All replies were included in the analysis. This is stated in the methods section.
- The introduction does not deal with the serotype specific protection TAK003 provides and does not point out this aspect of the vaccine. Its constitution, efficacy and safety are not well described and referenced.
A brief explanation regarding the details of TAK003 as life attenuated vaccine has been added. However, it is not the intention of this paper to give detailed information on TAK003. This has been done elsewhere, as cited in our paper. We feel that a detailed description of constitution, efficacy and safety of TAK003 would exceed the scope of this paper.
- Table 2 and 3 could have been made into figures to represent the data better.
Tables 2 and 3 offer detailed infromation on effects of co-vaccination vs. single vacination and adverse events in age groups. We feel that these details are better presented in tabels than in figures.
- Some statistical analysis and confidence values could have been provided to better understand the data, all in terms of days of symptoms, their range differences in age and symptoms due to co-vaccination.
As stated above, statistical methods were applied where feasible, significant differences are shown throughout the paper.
- The authors should have gotten more information on the individuals that got dengue after vaccination. Although, its understandable that the study design does not allow to do so, but it was valuable information which should not have been ignored.
We fully agree that that it would be useful to have more information on the individuals that got dengue after vaccination and in fact on all subjects who reported to the quetsionnaire. However, the design of anonymised data reporting does not allow for further inquries. This is stated as one of the weaknesses of the study in the discussion.
Line 15-20 The information and numbers could be distilled further to make the abstract concise and crisp.
The abstract has been shortened, information on age median omitted.
Line 43 Please provide reference.
A reference has been added
Line 44 Please provide reference.
A reference has been added
Line 52-54 Please provide reference.
A reference has been added
Line 60 Seems like this is the only other vaccine that has been developed.
That is correct.
Line 69 No information about limited safety against DENV-3 and DENV-4 serotypes.
Ther are no data showing limited safety against DENV-3 and DENV-4 serotypes. Information on the limited efficacy against DENV-3 and DENV-4 serotypes in pivotal studies has been added.
Line 98 The ages are somewhere written as average as well as median and the values don’t match in Abstract and Results.
We apologise for this mistake. The absract has been corrected.
Line 116 Table 1 not described here but later, if that order is appropriate then change the order of the tables.
Frequency of adverse events are actually shown in table 1, order of tables is correct.
Line 233 The number of incidences of dengue post vaccination does not match the value written in abstract and results.
This has been corrected
Round 2
Reviewer 1 Report
Comments and Suggestions for Authors
Thank you for addressing my suggestions appropriately. I have no additional comments at this time.
Comments on the Quality of English LanguageWell written.
Reviewer 3 Report
Comments and Suggestions for Authors
No further comments are needed